# Bridging Debiasing Tasks with Sufficient Projection: A General Theoretical Framework for Vector Representations

## Abstract

Pre-trained vector representations in natural language processing often inadvertently encode undesirable social biases. Identifying and removing unwanted biased information from vector representation is an evolving and significant challenge. Our study uniquely addresses this issue from the perspective of statistical independence, proposing a framework for reducing bias by transforming vector representations to an unbiased subspace using sufficient projection. The key to our framework lies in its generality: it adeptly mitigates bias across both debiasing and fairness tasks, and across various vector representation types, including word embeddings and output representations of transformer models. Importantly, we establish the connection between debiasing and fairness, offering theoretical guarantees and elucidating our algorithm's efficacy. Through extensive evaluation of intrinsic and extrinsic metrics, our method achieves superior performance in bias reduction while maintaining high task performance, and offers superior computational efficiency.

## 1 Introduction

Natural Language Processing (NLP) models have made significant strides in recent years, with much of their success attributed to representation learning - the process of creating effective vector representations for textual data. Various research has been conducted in this area, including static word embedding (Mikolov et al., 2013; Pennington et al., 2014), contextualized embedding (Peters et al., 2018; Devlin et al., 2018; Radford et al., 2019), sentence embedding (Reimers and Gurevych, 2019) in addition to other representation methods.

However, as vector representations have been applied in a wide range of real-life scenarios, researchers have discovered that stereotypical biases and spurious correlations can be transferred from human-generated corpora to vector representations and models (Bolukbasi et al., 2016; Caliskan et al., 2017; Vig et al., 2020). This has the potential to produce biased and unfair outcomes in various downstream tasks (Kurita et al., 2019) and can even lead to serious social problems. For instance, in the word analogy task presented in (Bolukbasi et al., 2016), it was found that the vector representation for $\overrightarrow{she}$ was closer to $\overrightarrow{nurse}$ than the representation for $\overrightarrow{he}$ was to $\overrightarrow{doctor}$. De-Arteaga et al. (2019) found a performance gap between different genders in text classification tasks.

The bias and fairness issues in NLP models are primarily caused by the unbalanced and stereotypical nature of the training corpora. Liang et al. (2020) described this as unbalanced model behavior in relation to certain socially sensitive topics such as gender, race, and religion. To quantify biases in NLP, two types of bias evaluation metrics(intrinsic and extrinsic) have been proposed. However, recent research has shown that in most cases, there is a weak correlation between them (Goldfarb-Tarrant et al., 2021; Kaneko et al., 2022; Cao et al., 2022). There remains a significant research gap in understanding how to bridge these two kinds of tasks. In our research, we employ statistical independence to establish a theoretical linkage between these tasks, offering insights into the interplay between intrinsic and extrinsic biases.

Various methods have been proposed for reducing bias in NLP, but it remains a challenge to effectively mitigate bias while maintaining high model performance. Furthermore, it is particularly difficult

for debiasing methods to efficiently address both intrinsic and extrinsic biases at the same time, as discussed in the related works section.

In this paper, we propose a general debiasing method that can effectively mitigate bias across both debiasing and fairness tasks. Our key contributions include the following:

- We are the first to scope the debiasing and fairness tasks unitedly through statistical independence, providing theoretical analysis to bridge the connection between them.
- Our algorithm showcases its effectiveness in both intrinsic(bias embedding evaluation) and extrinsic(fairness text classification) evaluation metrics. It is versatile, adapting to different embedding methods and sensitive variable types, making it universally applicable.
- Our method improves upon existing state-of-the-art methods while still maintaining good task performance compared to the original model and stands out due to its superior computational efficiency.

The structure of this paper is as follows. We begin with a comprehensive review of existing research on bias evaluation and debiasing techniques in NLP. We then introduce our methodology, including our proposed algorithm. We present experimental results on a range of gender bias evaluation tasks, showcasing the effectiveness of our approach. Finally, we provide a theoretical bridge and guarantee, and a discussion of our method.

## 2 RELATED WORKS

**Debiasing Methods in NLP**    Researchers have been focusing on reducing bias from each component of NLP models. The most intuitive idea of debiasing is through counterfactual data augmentations (Zmigrod et al., 2019; Dinan et al., 2020; Barikeri et al., 2021), which involves re-balancing a corpus by swapping bias attribute words (e.g., he/she) in a dataset. The re-balanced corpus is then used for further training to debias a model. While this approach is simple and can be applied to all tasks, it does not perform well in terms of debiasing and requires additional computational resources for model re-training. Another direction is fine-tuning pre-trained transformer-based language models using methods such as projection (Kaneko and Bollegala, 2021), adversarial (Han et al., 2021), contrastive (Cheng et al., 2020; Shen et al., 2021; He et al., 2022), dropout (Webster et al., 2020), and prompting (Schick et al., 2021; Guo et al., 2022). These methods show effectiveness in reducing bias in various intrinsic evaluation tasks. However, when deploying these debiased models to downstream tasks, especially fine-tuning on task-specific datasets, the debiased language model can still re-learn social bias, making these debiasing methods less effective.

Our proposed debiasing method is based on the controlled removal of specific information from vector representations, which is closely related to the task of disentangling representations (Bengio et al., 2013). Previous research in this area includes methods for removing bias from static embeddings, such as projecting the word embedding into the orthogonal space of the gender direction ($\overrightarrow{he}$ - $\overrightarrow{she}$) (Bolukbasi et al., 2016), re-training the entire embedding using some loss functions (Kaneko and Bollegala, 2019), and utilizing the ideas in causal inference (Yang and Feng, 2020; Ding et al., 2022). There are several similar projection-based methods like Iterative Nullspace Linear Projection (INLP; (Ravfogel et al., 2020)), RLACE (Ravfogel et al., 2022a). We discuss the detailed comparison and advantages of our method in Section 7.

**Evaluating Bias in NLP**    The measurement methods for evaluating bias in pre-trained word embeddings and language models can be broadly divided into two categories: Intrinsic and Extrinsic evaluations. Intrinsic bias evaluations probe the bias within pre-trained word embeddings and language models. Common methods include measuring the geometry in embedding space, such as the Word Embedding Association Test (WEAT; Caliskan et al. (2017)) and Sentence Encoder Association Test (SEAT; May et al. (2019)). Additionally, Kurita et al. (2019); Nangia et al. (2020); Nadeem et al. (2021) propose metrics using the likelihood score. Furthermore, research suggests that some debiasing methods may only hide bias, and thus additional measurement approaches are needed Gonen and Goldberg (2019).

The extrinsic bias is specific to certain downstream tasks. In the text classification task, De-Arteaga et al. (2019); Blodgett et al. (2016) proposed two benchmark datasets and used the equal opportu-

nity measure from fairness literature. Zhao et al. (2018a) proposed the WinoBias benchmark for Coreference resolution. As well as other benchmarks, such as Bias-NLI (Dev et al., 2020) and in machine translation (Stanovsky et al., 2019). However, recent research has indicated that intrinsic bias in embeddings or models typically does not have a strong correlation with bias in downstream tasks(Goldfarb-Tarrant et al., 2021; Cao et al., 2022). Kaneko et al. (2022) found out that the debiased models re-learn the bias from the fine-tuning datasets, showing that only debiasing upstream models may not be enough to eliminate bias in downstream tasks.

In our work, we conduct comprehensive evaluation experiments in both intrinsic and extrinsic tasks. Additionally, our approach avoids the issue of re-learning bias by directly addressing the vector representation in downstream tasks.

## 3 METHODOLOGY

### 3.1 PROBLEM SETUP

We consider the problem of removing sensitive information inside the vector representation. Given the representation vector $X \in \mathbb{R}^{p_1}$ accompanied with the target attribute $Y \in \mathbb{R}^{p_2}$ and the sensitive attribute $Z \in \mathbb{R}^{p_3}$, our goal is to find a map $g : \mathbb{R}^{p_1} \mapsto \mathbb{R}^{p_1}$ such that:

- $g(X)$ is uncorrelated with $Z$;
- $g(X)$ maintains ability to predict $Y$.

In other words, the new representation $\widetilde{X} = g(X)$ removes the sensitive information $Z$ contained in the original representation while preserving other useful information in $X$. The notations given above incorporate debiasing and fairness tasks into the same framework, which are formulated in the following definitions:

**Definition 3.1** (Debias). *Let $\widetilde{X} = g(X)$ and $f_1$ be the model with input $\widetilde{X}$. Then $\widetilde{X}$ is said to be a debiased representation if $f_1(\widetilde{X}) \perp\!\!\!\perp Z$.*

**Definition 3.2** (Fairness). *Let $\widetilde{X} = g(X)$ and $f_2$ be the predictor trained by $(\widetilde{X}, Y)$. Then $\widetilde{X}$ is said to be a fair representation if $f_2(\widetilde{X}) \perp\!\!\!\perp Z \mid Y$.*

Our formulated definition directly aligns with the objectives of each task. In the debiasing task, given the input $X$, the model $f_1$ is biased if the output $f_1(X)$ relies heavily on $Z$. Ideally, an unbiased model should satisfy that the distribution of $Z$ is uniform over its support. Therefore, the goal is to make $Z$ and the output of the model $f_1(X)$ to be independent given the representation $X$. Similarly, for fairness tasks, the aim is to develop a fair predictor $f_2$, ensuring its prediction is independent of sensitive information $Z$ conditioned on $Y$ (Hardt et al., 2016).

### 3.2 MOTIVATION

As previously discussed, our goal is to identify a mapping $g$ such that $\widetilde{X} = g(X)$ possesses the desired properties. A direct approach involves constructing a $p_1 \times p_1$ projection matrix $P$ and applying the linear transformation $g(X) = PX$. When we restrict the transformation $g$ into linear projections, the original vector $X$ can be decomposed into $X = PX + (I - P)X$, where $I$ is the $p_1 \times p_1$ identity matrix. Letting $\mathcal{S}_1$ and $\mathcal{S}_2$ represent the spaces spanned by $P$ and $I - P$ respectively, the representation space is then decomposed as $\mathbb{R}^{p_1} = \mathcal{S}_1 \bigoplus \mathcal{S}_2$. For a debiased representation $\widetilde{X} = PX$, $\mathcal{S}_1$ should be structured to minimize the information regarding $Z$, while $\mathcal{S}_2$ should encapsulate as much of the information regarding $Z$ as possible.

Consequently, the debiasing goal transforms into the identification of the subspaces $\mathcal{S}_1$ and $\mathcal{S}_2$. It's crucial to note that the majority of information regarding $Z$ resides within $\mathcal{S}_2$. Therefore, the expression $(I - P)X$ emerges as a potential predictor for $Z$. Throughout this paper, we adhere to the following linearity assumption,

**Assumption 3.3** (Linearity). *$Z$ can be perfectly predicted by the linear combinations of $X$.*

With this assumption, denote $Q = I - P$ with its rank equal to $q$. If we suppose the orthogonal basis of $Q$ is $(\beta_1, \ldots, \beta_q)$, with each $\beta_j$ belonging to $\mathbb{R}^{p_1}$, then we can assume the following model as an

ideal representation where $\boldsymbol{Z}$ can be perfectly predicted by projecting $\boldsymbol{X}$ onto $q$ distinct directions:

$$\boldsymbol{Z} = f(\beta_1^\top \boldsymbol{X}, \dots, \beta_q^\top \boldsymbol{X}, \varepsilon), \tag{1}$$

where $f$ is an unknown function, which can be linear or nonlinear, and $\varepsilon$ denotes the random effect.

Based on the discussion above, the primary objective of this study is articulated as follows: initially, to identify the direction matrix $B = (\beta_1, \dots, \beta_q) \in \mathbb{R}^{p_1 \times q}$ ensuring that $B^\top \boldsymbol{X}$ captures most of the information about $\boldsymbol{Z}$. Then given this direction matrix, we can obtain a debiased representation of $\boldsymbol{X}$ by projecting it onto the subspace orthogonal to $\mathrm{Span}\{B\}$. Note that $\mathrm{Span}\{B\}$ is equivalent to $\mathrm{Span}\{Q\} = \mathcal{S}_2$, and hence, its corresponding orthogonal subspace $\mathcal{S}_1$ is spanned by $P$ as preliminarily defined. This approach underpins the theoretical foundation of our proposed algorithm, details of which will be explored in the forthcoming sections.

### 3.3 Minimal Subspace

It is crucial that we want the subspace $\mathcal{S}_2$ with the desired property as **small** as possible so that we can retain the utility of $\boldsymbol{X}$ after projecting it on $\mathcal{S}_1$. Essentially, we want to find the matrix $Q$ with minimum rank $q$. Consider the random variables $\boldsymbol{X} \in \mathbb{R}^{p_1}$ and $\boldsymbol{Z} \in \mathbb{R}^{p_3}$. If there exists a full rank matrix $B \in \mathbb{R}^{p_1 \times q}$, such that $\boldsymbol{Z} \perp\!\!\!\perp \boldsymbol{X} \mid B^\top \boldsymbol{X}$ ($\boldsymbol{X}$ is independent of $\boldsymbol{Z}$ conditioned on its projections on $B$), then the column space of the matrix $B$, denoted as $\mathrm{Span}\{B\}$, is called a sufficient dimension reduction subspace of $\boldsymbol{Z}$ with respect to $\boldsymbol{X}$. The intersection of all the dimension reduction subspaces is called the central subspace and denoted as $\mathcal{S}_{\boldsymbol{Z}|\boldsymbol{X}}$. That is $\mathcal{S}_{\boldsymbol{Z}|\boldsymbol{X}} = \bigcap_{B \in \mathcal{B}_{XZ}} \mathrm{Span}\{B\}$, where

$$\mathcal{B}_{XZ} = \left\{ B \mid \boldsymbol{Z} \perp\!\!\!\perp \boldsymbol{X} \text{ conditioned on } B^\top \boldsymbol{X} \right\}$$

The dimension of the central subspace is denoted as $\dim(\mathcal{S}_{\boldsymbol{Z}|\boldsymbol{X}})$. When $\mathrm{Span}\{B\}$ is the central subspace, we have $\dim(\mathcal{S}_{\boldsymbol{Z}|\boldsymbol{X}}) = \dim(\mathrm{Span}\{B\}) = q$. See Cook and Li (2002) for more details.

If $\boldsymbol{Z} \perp\!\!\!\perp \boldsymbol{X} \mid B^\top \boldsymbol{X}$, then $B^\top \boldsymbol{X}$ are most useful to predict $\boldsymbol{Z}$ based on $\boldsymbol{X}$, which is exactly the case in Model (1). Therefore, the central subspace $\mathcal{S}_{\boldsymbol{Z}|\boldsymbol{X}}$ is the minimal subspace processing the desired property and serves as a promising candidate for the expected subspace $\mathcal{S}_2$. We will illustrate the estimation procedure in the next section, which is robust to any kind of mapping $f$ in Model (1) and only relies on the data set $\{(X_i, Z_i)\}_{i=1}^n$.

Note that Model (1) is an ideal case based on the linearity assumption. In real-world applications, using only $q$ directions might not be sufficient to cover all the information of $\boldsymbol{Z}$ because there might be nonlinear correlations between $\boldsymbol{X}$ and $\boldsymbol{Z}$. However, since the $q$ directions can cover most of the information, it is still safe to use Model (1) in practice. Specifically, in NLP tasks, we may assume that the sensitive attribute $\boldsymbol{Z}$ can be predicted by the projections of the representation $\boldsymbol{X}$ onto $q$ directions with some unknown mapping $f$.

## 4 Sufficient Universal Projection (SUP)

### 4.1 Subspace Estimation

In this subsection, we will explain the process of estimating the subspace $\mathcal{S}_2 = \mathcal{S}_{\boldsymbol{Z}|\boldsymbol{X}}$ in different scenarios. We will begin with the simplest case where $p_3 = 1$, meaning $\boldsymbol{Z} \in \mathbb{R}$ is a scalar. Sliced Inverse Regression (SIR) is a classical dimension reduction method proposed by Li (1991) for univariate response $\boldsymbol{Z}$. We provide the detailed scheme for SIR applied on the data set $\{(X_i, Z_i)\}_{i=1}^n$ in the Appendix A.1. The main procedure of SIR is: (1) divide the support of $\boldsymbol{Z}$ into $H$ intervals and calculate the covariance matrix of $\boldsymbol{X}$ for each interval, (2) calculate the weighted covariance matrix based on $H$ intervals, (3) obtain the directions from the weighted covariance matrix. The space spanned by the directions $B = (\beta_1, \dots, \beta_q)$ provided by SIR is a consistent estimation of $\mathcal{S}_{\boldsymbol{Z}|\boldsymbol{X}}$.

For multivariate sensitive attributes, a direct analogy of the slicing strategy in SIR no longer works, as the number of partitions of the support of $\boldsymbol{Z} = (\boldsymbol{Z}_1, \dots, \boldsymbol{Z}_{p_3}) \in \mathbb{R}^{p_3}$ becomes $H^{p_3}$ and thus suffers the curse of dimensionality. To address the limitation of the original SIR, the Pooled Marginal Slicing (PMS) estimator proposed in Aragon (1997) combines the subspaces $\mathcal{S}_{\boldsymbol{Z}_i|\boldsymbol{X}}$ estimated by univariate response SIR to get the directions for the multivariate response, which is motivated by the following proposition.

**Proposition 4.1.** *Note that $\boldsymbol{Z} \perp\!\!\!\perp \boldsymbol{X} \mid B^\top \boldsymbol{X}$ implies $\boldsymbol{Z}_j \perp\!\!\!\perp \boldsymbol{X} \mid B^\top \boldsymbol{X}$. Therefore, for $j = 1, \ldots, p_3$, $\mathcal{S}_{\boldsymbol{Z}_j \mid \boldsymbol{X}} \subseteq \mathcal{S}_{\boldsymbol{Z} \mid \boldsymbol{X}}$.*

*Proof.* For any $B \in \mathcal{B}_{XZ}$, we have $B \in \mathcal{B}_{XZ_j}$, thus $\mathcal{B}_{XZ} \subset \mathcal{B}_{XZ_j}$. Recall that $\mathcal{S}_{\boldsymbol{Z} \mid \boldsymbol{X}}$ is the intersection of all elements in $\mathcal{B}_{XZ}$. Therefore, we have $\mathcal{S}_{\boldsymbol{Z}_j \mid \boldsymbol{X}} \subseteq \mathcal{S}_{\boldsymbol{Z} \mid \boldsymbol{X}}$ for $j = 1, \ldots, p_3$. $\square$

Proposition 4.1 indicates that $\mathcal{S}_{\boldsymbol{Z}_j \mid \boldsymbol{X}}$ can be used to recover $\mathcal{S}_{\boldsymbol{Z} \mid \boldsymbol{X}}$, which guarantees the theoretical properties of PMS estimator. It also naturally lifts the curse of dimensionality. Let $Z_{ij}$ denote the $j$-th coordinate of $i$-th sample. We apply SIR to data set $\{(X_i, Z_{ij})\}_{i=1}^n$ and obtain the estimators $M_i^{\mathrm{SIR}}$ for $j = 1, \ldots, p_3$. Then we define the weighted sum of estimators as $M^{\mathrm{PMS}} = \sum_{j=1}^{p_3} w_i M_i^{\mathrm{SIR}}$, where $w_i$ can be chosen as either equal weights or proportional to the leading eigenvalues of $M_i$. Then the leading $q$ eigenvectors $\psi_1, \ldots, \psi_q$ of $M^{\mathrm{PMS}}$ can be used to recover $\mathcal{S}_{\boldsymbol{Z} \mid \boldsymbol{X}}$. The detailed implementation of obtaining $M^{\mathrm{PMS}}$ is summarized in Algorithm 2 in the Appendix.

## 4.2 Algorithm Implementation

To obtain the debiased representation, we first collect the original vector representation $X_i$ and the associated sensitive attribute $Z_i$. Note that $Z_i$ can be labeled by humans or learned from the training data. Specifically, we place no restrictions on the structure of $Z_i$ – it can be either discrete labels $Z_i \in \{1, 2, \ldots, k\}$ representing gender or race, or a continuous variable. When $Z_i$ is a continuous variable, we directly set $\widetilde{Z}_i = Z_i$ and handle $\{(X_i, \widetilde{Z}_i)\}$ as discussed above. If $Z_i$ is a categorical variable with choices $\{1, 2, \ldots, k\}$, we first train a classifier $f_{\mathrm{cls}}$ based on the data set $\{(X_i, Z_i)\}_{i=1}^n$, whose output is the probability of $X_i$ belonging to each category, then denote

$$\widetilde{Z}_i = f_{\mathrm{cls}}(X_i) = (\widetilde{Z}_{i1}, \ldots, \widetilde{Z}_{ik}) \in \mathbb{R}^k,$$

where $\sum_{j=1}^k \widetilde{Z}_{ij} = 1$. In both scenarios, the attribute $Z_i$ is converted to the vector variable with continuous support. Then we can obtain the PMS estimator $M^{\mathrm{PMS}}$ based on $\{(X_i, \widetilde{Z}_i)\}_{i=1}^n$ with its leading $q$ eigenvectors $\psi_1, \ldots, \psi_q$. The projection matrix is defined as $P = I - \sum_{i=1}^q \psi_i \psi_i^\top$. Intuitively, $Q = \sum_{i=1}^q \psi_i \psi_i^\top$ is the estimated central mean space regarding $Z$, and the space spanned by this matrix contains most of the information we want to eliminate. The procedure is outlined in Algorithm 1.

---

**Algorithm 1** Sufficient Universal Projection (SUP)

**Input:** Data $\{(X_i, Z_i)\}_{i=1}^n$, partition $H$ and number of dimension $q$;
**Output:** Sufficient projection $P$;

 1: **if** $Z_i$ is continuous **then**
 2:     Set $\widetilde{Z}_i = Z_i$;
 3: **else if** $Z_i$ is discrete **then**
 4:     Train a classifier $f_{\mathrm{cls}}$ by $\{(X_i, Z_i)\}_{i=1}^n$;
 5:     Set $\widetilde{Z}_i = f_{\mathrm{cls}}(X_i)$;
 6: **end if**
 7: Obtain $M^{\mathrm{PMS}}$ using $\{(X_i, \widetilde{Z}_i)\}_{i=1}^n$ by Algorithm 2;
 8: Calculate the leading eigenvectors $\{\psi_j\}_{j=1}^q$ of $M^{\mathrm{PMS}}$;
 9: Obtain $P = I_{p_1} - \sum_{j=1}^q \psi_j \psi_j^\top$;
10: **Return**: $P$.

---

It is worth emphasizing that SUP is a general framework for bias elimination, and we have no assumption on the type of representation. Therefore, our proposed method is universally robust to both static and contextualized embeddings with different dimensions and can be applied to any downstream tasks.

## 5 EXPERIMENT AND SETTINGS

### 5.1 STATIC WORD EMBEDDING EVALUATION TASKS

We begin by demonstrating our method in the context of debiasing static word embeddings using 300-dimensional GloVe embeddings (Pennington et al., 2014) pre-trained on English Wikipedia data. We first split all the words evenly into two classes by calculating the cosine similarity between each word embedding with the gender direction $\overrightarrow{he}$ - $\overrightarrow{she}$. The label of each class is the sensitive attribute $Z$ and the projection matrix $P$ is calculated through Algorithm 1. For a fair comparison, the following evaluations are based on the methodology outlined in (Gonen and Goldberg, 2019; Ding et al., 2022). We compare our results with the following baseline methods: hard-debiasing method (Hard) (Bolukbasi et al., 2016), gender-preserving debiasing method (GP) (Kaneko and Bollegala, 2019), word vector learning method (GN) (Zhao et al., 2018b), half-sibling regression (HSR) (Yang and Feng, 2020), INLP (Ravfogel et al., 2020) and DeSIP (Ding et al., 2022).

**Clustering Gender Biased Words.**    Biased words tend to cluster together, and debiased embeddings may not escape this phenomenon. We use K-means clustering (K=2) to split the top 500 male-biased and top 500 female-biased words. A visualization graph is presented in Appendix A.3. In Table 1 column one, we report the accuracy in splitting the 1,000 words into male and female clusters. Our method brings about a 50% reduction compared with the original GloVe and about 20% compared with the second-best method.

**Correlation**    Using the top 50,000 most frequent words as targets, we calculate the Pearson correlation coefficient between the bias-by-projection and bias-by-neighbor results. The latter is calculated using the neighborhood metric, which counts the percentage of male and female-biased words within the 100 nearest neighbors of each target word. The result is presented in the second column of Table 1, and we achieve the lowest correlation coefficient.

**Profession Words**    In this task, we determine the correlation between the original bias and the number of male neighbors among the 100 nearest neighbors of profession words, as listed by Bolukbasi et al. (2016); Zhao et al. (2018b). The correlation coefficient is shown in Table 1. Our method reduces the coefficient by 20% compared with the original GloVe and achieves the best result.

**Classifcation**    We selected the top 2,500 biased words for each gender and trained a support vector machine (SVM) model using 1,000 randomly sampled words for each baseline model. We then applied the trained classifier to the remaining 4,000 words to predict gender bias direction. The prediction accuracy is shown in Table 1. Lower accuracy implies that the original embedding does not contain enough gender-related information. Our method has the least accuracy among all debiasing methods, indicating that it preserves the least gender bias.

|       | Clustering | Correlation | Profession | Classify |
|-------|-----------|-------------|------------|----------|
| GloVe | 1.0000    | 0.7727      | 0.8200     | 0.9980   |
| Hard  | 0.8050    | 0.6884      | 0.7161     | 0.9068   |
| GP    | 1.0000    | 0.7700      | 0.8102     | 0.9978   |
| GN    | 0.8560    | 0.7336      | 0.7925     | 0.9815   |
| HSR   | 0.9410    | 0.6422      | 0.6804     | 0.9055   |
| INLP  | 0.6336    | 0.5718      | 0.6651     | 0.8160   |
| DeSIP | 0.7920    | 0.6421      | 0.7060     | 0.8550   |
| **SUP** | **0.5198** | **0.5360** | **0.6515** | **0.7247** |

|       | Task1 | | Task2 | |
|-------|-------|-------|-------|-------|
|       | $p$ ($\uparrow$) | $d$ ($\downarrow$) | $p$ ($\uparrow$) | $d$ ($\downarrow$) |
| GloVe | 0.090 | 0.704 | 0.00* | 1.905 |
| Hard  | 0.363 | 0.187 | 0.00* | 1.688 |
| GP    | 0.055 | 0.832 | 0.00* | 1.909 |
| GN    | 0.157 | 0.541 | 0.074 | 0.753 |
| HSR   | 0.265 | 0.340 | 0.00* | 1.555 |
| INLP  | 0.195 | 0.475 | 0.129 | 0.595 |
| DeSIP | 0.268 | 0.335 | 0.001* | 1.462 |
| **SUP** | **0.411** | **0.119** | **0.142** | **0.565** |

Table 1: **Left:** Static word embedding bias evaluation tasks. A lower number in each column indicates better debiasing performance. Baseline results are reported by Ding et al. (2022). Our method surpasses all other methods; **Right:** WEAT result. In each column of $p$-value, * indicates statistically significant compared with $\alpha = 0.05$; In each column of $d$, a value closer to 0 is indicative of less bias. The best results are boldfaced.

**Word Embedding Association Test**    The WEAT (Caliskan et al., 2017) is a permutation-based test that measures bias in word embeddings. Please refer to Appendix A.4 for the details of WEAT.

The results are reported in terms of absolute effect sizes($d$) and $p$-values ($p$). The effect size is a normalized measure of how separated two distributions are. A high effect size indicates a larger bias between the target and attribute words, and the $p$-value denotes whether the bias is statistically significant or not. We conduct two tests using the Pleasant & Unpleasant (Task 1) and Career & Family (Task 2) word sets. We consider male and female names as attribute sets. As shown in Table 1, in both tasks, the $p$-value is not significant, indicating the bias is non-significant. We also achieve the smallest effect size in both of the tasks, indicating the effectiveness in reducing bias in word embeddings.

## 5.2 Word Similarity Tasks

While reducing bias is our primary goal, it is crucial not to destroy other semantic information encoded in word embeddings. We evaluate our algorithm by the following word similarity tests: RG65 (Rubenstein and Goodenough, 1965), WordSim-353 (Finkelstein et al., 2001), Rarewords (Luong et al., 2013), MEN (Bruni et al., 2014), MTurk-287 (Radinsky et al., 2011), and MTurk-771 (Halawi et al., 2012), SimLex-999 (Hill et al., 2015), and SimVerb-3500 (Gerz et al., 2016). These datasets associated with each task contain word pairs and a corresponding human-annotated similarity score. We calculate Spearman's rank correlation coefficient between the two ranks. The results of our method and the original GloVe are shown in Table 2. We observe an overall non-decreasing performance in most of the tasks, showing that the semantic information is protected.

|  | RG65 | WS | RW | MEN |
|---|---|---|---|---|
| GloVe | 0.7540 | 0.6199 | 0.3722 | 0.7216 |
| **SUP** | **0.7913** | **0.6617** | **0.3986** | **0.7423** |
|  | MT-287 | MT-771 | SimLex | SimVerb |
| GloVe | 0.6480 | 0.6486 | 0.3474 | 0.2038 |
| **SUP** | 0.6349 | **0.6792** | **0.3949** | **0.2493** |

|  | Gender($\downarrow$) | Race ($\downarrow$) | Religion ($\downarrow$) |
|---|---|---|---|
| BERT | 0.620 | 0.620 | 0.492 |
| +CDA | 0.722 | 0.569 | 0.339 |
| +Dropout | 0.765 | 0.554 | 0.377 |
| +INLP | **0.204** | 0.639 | 0.460 |
| +SentDebias | 0.434 | 0.612 | 0.439 |
| **+SUP** | 0.218 | **0.432** | **0.261** |

Table 2: **Left:** Word similarity results. A higher value indicates a better semantic correlation; **Right:** SEAT average effect sizes for debiased BERT. A lower number in each column indicates better debiasing performance. The best results are boldfaced. Baseline results are from Meade et al. (2022).

## 5.3 Sentence Embedding Association Test (SEAT)

In addition to testing on static word embeddings, we also test on contextualized word embeddings. SEAT (May et al., 2019), extends the WEAT test by leveraging simple templates such as 'This is a <word>' to obtain the individual word's contextualized embedding. We use the implementation results from (Meade et al., 2022). The baseline includes BERT base uncased, CDA and Dropout (Webster et al., 2020), SentDebias (Liang et al., 2020), and INLP.

For a detailed list of the SEAT tests used to measure each type of bias in our work, the complete results, as well as obtaining the projection, we refer readers to Appendix A.5. In Table 2, we display the average effect size for each SEAT task category evaluated. Our findings reveal superior performance in two out of the three tasks while delivering comparable results to the INLP method in the Gender task. Notably, our SUP method exhibits enhanced performance across a variety of bias-influenced topics.

## 5.4 Extrinsic: Fairness Text Classification

For the extrinsic task, we consider the fairness text classification problem. We conduct experiments over three different tasks – sentiment analysis (MOJI) Blodgett et al. (2016), biography classification (BIOS) De-Arteaga et al. (2019). The detail of the datasets is described in Appendix A.6.

The fairness criterion is defined by *Equality of Opportunity* (EO), i.e. a classifier is considered fair if its prediction is independent of the sensitive attribute given the true label. For BIOS and MOJI data, it is measured by considering the gap in the True Positive Rate (TPR) between different sensitive attribute groups:

$$TPR_{z,y} = P[\hat{Y} = y | Z = z, Y = y], \quad GAP_y^{TPR} = TPR_{z,y} - TPR_{z',y}$$

The root-mean-square (RMS) gap over all groups is $GAP_{RMS}^{TPR} = \sqrt{\frac{1}{|C|}\sum_{y \in C}(GAP_y^{TPR})^2}$.

We follow the original implementation of MOJI and BIOS that use race and gender labels as sensitive attributes $Z$. The results are shown in Table 3. We report the Accuracy, the $GAP_{RMS}$, and the Time in seconds for BIOS and MOJI. The baselines are from (Ravfogel et al., 2020), (Ravfogel et al., 2022a), (Chowdhury and Chaturvedi, 2022), and (Ravfogel et al., 2022b).

| | BIOS | | | MOJI | | |
|---|---|---|---|---|---|---|
| | Acc.($\uparrow$) | GAP($\downarrow$) | Time($\downarrow$) | Acc.($\uparrow$) | GAP($\downarrow$) | Time($\downarrow$) |
| BERT | 79.1 | 14.5 | - | 71.6 | 31.0 | - |
| +INLP | 71.9 | 9.9 | 271 | 62.2 | 15.8 | 1003 |
| +RLACE | 76.9 | 13.2 | 4312 | 72.2 | 15.4 | 2456 |
| +FaRM(unconstrained) | 55 | **7.9** | 6723 | 63.5 | 14.0 | 4162 |
| +Kernel(Poly) | **79.9** | 16.8 | 3914 | 72.9 | 17.3 | 8861 |
| +Kernel(RBF) | 60.7 | 18.0 | 3487 | **74.1** | 13.3 | 5496 |
| **+SUP** | 76.4 | 12.7 | **6.76** | 69.1 | **10.5** | **33.04** |

Table 3: **Left:** Result of BIOS text classification. Predict using [CLS] token. **Right:** Result of MOJI text classification. The best result is boldfaced.

In Table 3, we implement each task using BERT and establish it as the baseline - this represents the results without any fairness considerations. Our findings reveal that in the BIOS task, while the INLP and FaRM achieve low RMS, it is accompanied by a compromise in accuracy. In contrast, our SUP method demonstrates balanced performance on both fronts. For MOJI, our algorithm stands out, yielding the smallest discrepancy gap among all methods, all the while maintaining uncompromised accuracy. Moreover, our algorithm benefits from having an explicit solution, eliminating the need for iterative calculations, and running significantly faster than many existing baselines.

In addition, we also conduct experiments on a more challenging dataset: the Toxic Comment Classification (Dixon et al., 2018). Within this dataset, each sample may belong to **multiple sensitive attribute groups**, embodying intersectionality in biases. For instance, a single comment might simultaneously belong to 'black' and 'gay' sensitive groups. We adhere to the definitions and gap measurements outlined by Dixon et al. (2018), $GAP_{\text{toxic}} = \sum_{z \in \mathcal{Z}} |TPR_{z,0} - \text{mean}_{z \in \mathcal{Z}}(TPR_{z,0})|$, where $\text{mean}_{z \in \mathcal{Z}}(TPR_{z,0})$ is the average of TPR gaps of all sensitive attributes. where $\text{mean}_{z \in \mathcal{Z}}(TPR_{z,0})$ represents the average of True Positive Rate (TPR) gaps across all sensitive attributes. The sensitive attribute in this scenario is depicted as a 50-dimensional vector, illustrating the relative frequency of sensitive words within sentences.

For the original BERT model, the Area Under the Curve (AUC) was 95.5, and the $GAP_{\text{toxic}}$ was 7.34. By employing our method, we managed to maintain the AUC at 95.0 while reducing the $GAP_{\text{toxic}}$ to 5.95, showcasing the efficacy of our approach in mitigating biases while preserving model performance. It is crucial to highlight that our methodology effectively manages the intricacies of intersectional biases in toxic comment classification, a complexity not adequately addressed by other baseline algorithms. For a more detailed discussion, please refer to Section 7 below.

## 6 BRIDGE BETWEEN DEBIASING AND FAIRNESS

In this section, we provide a theoretical analysis of how our proposed method can incorporate debiasing and fairness tasks into a unified framework and handle them simultaneously. As previously discussed, both tasks aim to achieve conditional independence with respect to certain variables. Here, we will demonstrate how minimal subspace $\mathcal{S}_2$ bridges these tasks. We provide the following theorem to prove the effectiveness of our framework in dealing with both debiasing and fairness tasks. For detailed proof, Please refer to Appendix A.7.

**Theorem 6.1.** *With the settings defined in Section 3 and linearity assumption, suppose $\boldsymbol{X} \perp\!\!\!\perp \boldsymbol{Z} \mid Q\boldsymbol{X}$, then $\widetilde{\boldsymbol{X}} = (I - Q)\boldsymbol{X}$ is a debiased representation. Further, suppose $\boldsymbol{X} \perp\!\!\!\perp \boldsymbol{Y} \mid Q_y\boldsymbol{X}$, if $\mathrm{Span}\{Q_y\} \subseteq \mathrm{Span}\{Q\}$, then $\widetilde{\boldsymbol{X}} = (I - Q)\boldsymbol{X}$ is a fair representation.*

Theorem 6.1 states that the projected representation $\widetilde{\boldsymbol{X}} = (I - Q)\boldsymbol{X}$ has no correlation with the sensitive attributes $\boldsymbol{Z}$, which achieves the goal of debiasing task. Moreover, if the subspace spanned

by sensitive attribute $Z$ ($\mathcal{S}_{Z|X}$) is included in the subspace spanned by target attribute $Y$ ($\mathcal{S}_{Y|X}$), we can achieve the goal of fairness task by projecting the original representation on $(I - Q)$.

The theoretical property is consistent with the experimental results shown above. For the debiasing task, the matrix $Q = \sum_{i=1}^{q} \psi_i \psi_i^{\top}$ in Algorithm 1 is the estimated central mean space regarding $Z$, then $I - Q$ forms a sufficient projection defined in Theorem 6.1, which shows great improvement upon existing state-of-the-art methods. For fairness task, the eigenvectors $\{\psi_j\}_{j=1}^{q}$ calculated in Algorithm 1 recovers the matrix $Q$ stated in Theorem 6.1. If we have $\mathrm{Span}\{Q_y\} \subseteq \mathrm{Span}\{Q\}$, then we can set $\widetilde{X} = (I - Q)X$ to get the fair representation. However, in real data, this condition is usually violated, which means $\mathrm{Span}\{Q_y\} \not\subseteq \mathrm{Span}\{Q\}$. Therefore, the SUP may not achieve the optimal fair representation in downstream tasks.

## 7 COMPARISON WITH OTHER PROJECTION METHODS

We conduct a comparative analysis between our method and other projection-based methods.

**INLP:** Both our method SUP and INLP employ linear projections to minimize the influence of the sensitive attribute $Z$ in the representations. The underlying principle of INLP revolves around identifying the null space of the weight matrix, denoted as $W \in \mathbb{R}^{k \times p_1}$, which corresponds to the parameters of linear classifier $Z = f(WX)$, where $f$ represents the classifier function and $k$ is the number of classes. This framework can be viewed as a specific instance of the Model 1, where the subspace spanned by $\beta_1, \ldots, \beta_q$ exactly corresponds to the union of row spaces of $W_i$ during iterations. Specifically, INLP captures $k$ directions (rows of weight matrix $W_j$) at each iteration, while SUP finds $q$ directions in a single run, which is more flexible and computationally efficient.

**RLACE**(Ravfogel et al., 2022a): Both SUP and RLACE operate under linearity assumption as expressed in Assumption 3.3. In RLACE, the function $f$ in Model (1) is interpreted as the inverse of a link function in the generalized linear model. In contrast, our approach imposes no specific constraints on the form of $f$. While RLACE achieves debiasing by solving a minimax problem to identify the projection $P$ that safeguards the sensitive attribute, our method directly estimates the directions with a closed form, offering superior computational efficiency.

**Advantage of SUP:** The main distinction between our proposed methodology and existing projection-based debiasing methods pertains to the range of tasks they can address. For instance, INLP is principally designed for handling categorical sensitive attributes. In the context of the toxic data task (see A.6), the sensitive attribute is no longer a categorical variable, thereby undermining the effectiveness of INLP. However, it is important to note that our SUP algorithm does not violate the structure of Model (1) under the linearity assumption. As a result, our approach remains capable of estimating the directions $\beta_1, \ldots, \beta_q$ and mitigating bias through the Algorithm 1. This highlights the versatility of our SUP algorithm, showcasing its capability to adeptly manage a spectrum of uni-/multivariate and discrete/continuous sensitive datasets. The capability of managing a sensitive attribute as a continuous variable also aligns more closely with contemporary sociological understandings. For instance, consider the interpretation of gender as a spectrum(Richards et al., 2016) rather than a binary categorization. As such, models that can accommodate continuous variables for sensitive attributes are better equipped to reflect these more nuanced perspectives, thereby promoting fairness and inclusivity in their outcomes.

## 8 CONCLUSION AND FUTURE WORK

In this paper, we propose a theoretically grounded framework for reducing bias by projecting vector representations to an unbiased subspace. It can reduce biased information effectively in both intrinsic and extrinsic tasks, as well as different kinds of representations. In addition, we provide a theoretical guarantee about the effectiveness of our method in reducing biased information.

While this work has demonstrated its effectiveness in various tasks, it has the potential to be applied to other applications that rely on vector representation. We are also interested in combining our method with the different other notions of fairness. We aim to explore these directions in future work.

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

# A APPENDIX

## A.1 SCHEME FOR SIR ESTIMATOR

Suppose the data set $\{(X_i, Z_i)\}_{i=1}^n$ is given, then the steps for SIR are summarized as :

1. Standardizing $\boldsymbol{X}$ by the transformation $\tilde{\boldsymbol{X}}_i = C_X^{-1/2}(\boldsymbol{X}_i - \mu_X)$, where $\mu_X$ and $C_X$ are the mean vector and covariance matrix of $X$.

2. Slice the range of $Z$ into $H$ intervals $\{J_h\}_{h=1}^H$. Estimate the weight $p_h = (1/n)\sum_{i=1}^n I(Z_i \in J_h)$ and compute the sample mean $m_h = (1/np_h)\sum_{Z_i \in J_h} \tilde{\boldsymbol{X}}_i$ on each sliced interval.

3. Form $M^{\text{SIR}} = \sum_{h=1}^H p_h m_h m_h^\top$ and let $\phi_k$ be its eigenvectors. The directions are estimated by $\beta_k = C_X^{-1/2}\phi_k$ for $k = 1, \ldots, q$.

## A.2 PMS ESTIMATOR IMPLEMENTATION

For multivariate variable $Z \in \mathbb{R}^{p_3}$, let $Z_{ij}$ denote the $j$-th coordinate of $i$-th sample, the PMS estimator can be achieved through the following Algorithm 2.

---

**Algorithm 2** PMS Estimator

---

**Input:** Data $\{(X_i, Z_i)\}_{i=1}^n$, partition $H$, covariance matrix $C_X$ and weights $\{w_j\}_{j=1}^{p_3}$;
**Output:** PMS estimator $M^{\text{PMS}}$;

1: **for** $j = 1, \ldots, p_3$ **do**
2:     Slice the support of $Z_j$ into $H$ intervals denoted as $\{J_{j,h}\}_{h=1}^H$
3:     **for** $h = 1, \ldots, H$ **do**
4:         Estimate the weight on each interval $p_{j,h} = \frac{1}{n}\sum_{i=1}^n I(Z_{ij} \in J_{j,h})$;
5:         Compute the standardized mean on each interval $m_{j,h} = \frac{1}{np_{j,h}}\sum_{Z_{ij} \in J_{j,h}} C_X^{-1}X_i$;
6:     **end for**
7:     Obtain the estimator for each dimension $M_j^{\text{SIR}} = \sum_{h=1}^H p_{j,h} m_{j,h} m_{j,h}^\top$;
8: **end for**
9: Calculate the weighted sum of estimators $M^{\text{PMS}} = \sum_{j=1}^{p_3} w_j M_j^{\text{SIR}}$;
10: **Return**: $M^{\text{PMS}}$.

---

The weights $w_j$ can be chosen as either equal weights or proportional to the leading eigenvalues of $M_j$. Then the leading $q$ eigenvectors $\psi_1, \ldots, \psi_q$ of $M^{\text{PMS}}$ can be used to recover $\mathcal{S}_{\boldsymbol{Z}|\boldsymbol{X}}$.

## A.3 T-SNE VISUALIZATION

To visually demonstrate the effectiveness of our proposed method in reducing gender bias, we selected the top 500 male- and female-biased embeddings. Using t-SNE projection, we generated a graph for the original GloVe and our debiased embeddings. Figure 1 shows the separation of male- and female-biased embeddings in two different colors. It can be observed that our method has mixed the male- and female-biased embeddings effectively.

## A.4 DETAIL OF WEAT

Let $X$ and $Y$ be two sets of target words of equal size $n$ with their embedding $\{x_i\}_{i=1}^n$ and $\{y_i\}_{i=1}^n$, and $A$, $B$ the two sets of attribute words with their embedding $\{a_i\}_{i=1}^{|A|}$ and $\{b_i\}_{i=1}^{|B|}$. The WEAT uses the difference of averaged distance to measure the similarity of a vector $w$ to two sets $A$ and $B$. The test statistic is

$$s(X, Y, A, B) = \sum_{x \in X} s(x, A, B) - \sum_{y \in Y} s(y, A, B)$$

where

$$s(w, A, B) = \frac{1}{|A|}\sum_{a \in A} \cos(w, a) - \frac{1}{|B|}\sum_{b \in B} \cos(w, b)$$

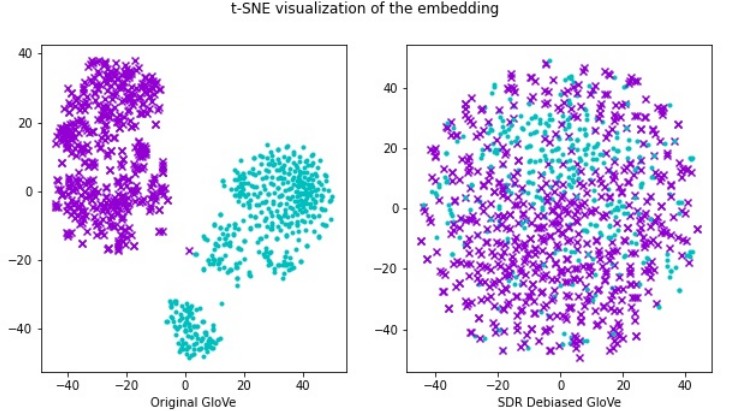

Figure 1: t-SNE visualization.

In other words, $s(w, A, B)$ measures the association of the word $w$ with the attribute, and $s(X, Y, A, B)$ measures the differential association of the two sets of target words with the attribute.

Let $\{(X_i, Y_i)\}_i$ denote all the partitions of $X \cup Y$ into two sets of equal size. The one-sided $p$-value of the permutation test is

$$\Pr_i \left[ s\left( X_i, Y_i, A, B \right) > s(X, Y, A, B) \right]$$

The effect size is

$$\frac{\text{mean}_{x \in X}\, s(x, A, B) - \text{mean}_{y \in Y}\, s(y, A, B)}{\text{std-dev}_{w \in X \cup Y}\, s(w, A, B)}$$

It is a normalized measure of how separated the two distributions (of associations between the target and attribute) are.

All word lists are from Caliskan et al. (2017). Because GloVe embeddings are uncased, we use lowercase words.

### A.5 DETAIL OF SEAT

#### A.5.1 OBTAIN THE PROJECTION MATRIX

To train projections for the topics of gender, race, and religion, we used the vocabulary from the GloVe model. All words were divided into groups according to their cosine similarities with pre-determined hint words: [he, she] for gender, [black people, white people] for race, and [Christianity, Jewish, Islam] for religion. Using BERT representations, we selected the top 75k words for gender, 75k for race, and 30k for religion from each group and associated them with their group labels as the input dataset for Algorithm 1.

#### A.5.2 FULL TEST AND RESULTS OF SEAT

In this section, we provide a complete set of results for all SEAT tests. All of the baseline results are from Meade et al. (2022). Also, for detailed attribute word sets and the target word sets, please refer to their GitHub repo. Table 4 are tasks for Gender debias. Table 5 are tasks for Race debias. Table 6 are tasks for Religion debias.

### A.6 FAIR TEXT CLASSIFICATION DETAILS

The **MOJI** is a sentiment classification dataset collected by Blodgett et al. (2016) that contains tweets from either African-American English or Standard American English. Each of the text data is labeled with a binary 'race' label based on the kind of English they use. The binary sentiment

| SEAT Gender Tasks | | | | | | | |
|---|---|---|---|---|---|---|---|
| Model | SEAT-6 | SEAT-6b | SEAT-7 | SEAT-7b | SEAT-8 | SEAT-8b | Avg. Effect Size ($\downarrow$) |
| BERT | 0.931 | 0.090 | -0.124 | 0.937 | 0.783 | 0.858 | 0.620 |
| CDA | 0.846 | 0.186 | -0.278 | 1.342 | 0.831 | 0.849 | 0.722 |
| Dropout | 1.136 | 0.317 | 0.138 | 1.179 | 0.879 | 0.939 | 0.765 |
| INLP | 0.317 | -0.354 | -0.258 | 0.105 | 0.187 | -0.004 | 0.204 |
| SentDebias | 0.350 | -0.298 | -0.626 | 0.458 | 0.413 | 0.462 | 0.434 |
| SUP | -0.028 | -0.286 | -0.403 | -0.255 | 0.213 | -0.124 | 0.218 |

Table 4: SEAT effect sizes for gender debiased BERT. Effect sizes closer to 0 are indicative of less biased model representations.

| SEAT Race Tasks | | | | | | | |
|---|---|---|---|---|---|---|---|
| Model | ABW-1 | ABW-2 | SEAT-3 | SEAT-3b | SEAT-4 | SEAT-5 | SEAT-5b | Avg. Effect Size ($\downarrow$) |
| BERT | -0.079 | 0.690 | 0.778 | 0.469 | 0.901 | 0.887 | 0.539 | 0.620 |
| CDA | 0.231 | 0.619 | 0.824 | 0.510 | 0.896 | 0.418 | 0.486 | 0.569 |
| Dropout | 0.415 | 0.690 | 0.698 | 0.476 | 0.683 | 0.417 | 0.495 | 0.554 |
| INLP | 0.295 | 0.565 | 0.799 | 0.370 | 0.976 | 1.039 | 0.432 | 0.639 |
| SentDebias | -0.067 | 0.684 | 0.776 | 0.451 | 0.902 | 0.891 | 0.513 | 0.612 |
| SUP | 0.019 | 0.428 | 0.542 | 0.193 | 0.611 | 0.716 | 0.514 | **0.432** |

Table 5: SEAT effect sizes for race debiased BERT. Effect sizes closer to 0 are indicative of less biased model representations.

| SEAT Religion Tasks | | | | | |
|---|---|---|---|---|---|
| Model | Religion-1 | Religion-1b | Religion-2 | Religion-2b | Avg. Effect Size ($\downarrow$) |
| BERT | 0.744 | -0.067 | 1.009 | -0.147 | 0.492 |
| CDA | 0.355 | -0.104 | 0.424 | -0.474 | 0.339 |
| Dropout | 0.535 | 0.109 | 0.436 | -0.428 | 0.377 |
| INLP | 0.473 | -0.301 | 0.787 | -0.280 | 0.460 |
| SentDebias | 0.728 | 0.003 | 0.985 | 0.038 | 0.439 |
| SUP | 0.392 | -0.066 | 0.492 | 0.092 | **0.261** |

Table 6: SEAT effect sizes for religion debiased BERT. Effect sizes closer to 0 are indicative of less biased model representations.

score is annotated by the emoji contained in the tweets. We compose the training data set as follows: AAE–happy = 40%, SAE– happy = 10%, AAE–sad = 10%, and SAE–sad = 40%. We used the train, dev, and test splits of 100k/8k/8k instances, respectively.

The **BIOS** dataset De-Arteaga et al. (2019) is a personal biography classification dataset annotated by gender and 28 classes of occupation. We follow the same split setup for the BIOS data as in De-Arteaga et al. (2019), and the ratio of train:dev:test is $65\% : 10\% : 25\%$.

The **Toxic** dataset features text sourced from the Talk Pages of Wikipedia, where each comment has been categorized by human assessors as either toxic or non-toxic. Interestingly, an analysis of this dataset has revealed a disproportionate appearance of certain demographic identity-related terms (such as "gay" and "black") within the labels. This imbalance can inadvertently lead to biased model training, resulting in discriminatory behavior towards certain groups. Our research employs the same division of data as specified by (Dixon et al., 2018), enabling us to test the efficacy of our method in reducing discrimination against minority groups.

### A.7 PROOF OF THEOREM 6.1

*Proof.* According to the definition of conditional independence, for any measurable function $f$, we have $f(X) \perp\!\!\!\perp Z \mid QX$ because the randomness of $f(X)$ only comes from the random variable $X$.

For the debias task, notice that $X \perp\!\!\!\perp Z \mid QX$, thus $X \perp\!\!\!\perp Z \mid QX$. It implies that $Z$ only depends on $QX$. Therefore, if we eliminate those correlated part and denote $\widetilde{X} = (I - Q)X$, we have $\widetilde{X} \perp\!\!\!\perp Z$. It achieves the goal of the debias task defined above.

For the fairness task, if we assume $X \perp\!\!\!\perp Y \mid Q_y X$, which implies $Y = f_0(Q_y X)$ for some measurable function $f_0$. Notice that $\mathrm{Span}\{Q_y\} \subset \mathrm{Span}\{Q\}$, then $\mathrm{Span}\{Q_y\}$ is orthogonal to $\mathrm{Span}\{I - Q\}$, which implies $(I - Q)X \perp\!\!\!\perp Q_y X$. Therefore, $(I - Q)X \perp\!\!\!\perp Z \mid Y$ since the randomness of $Y$ comes from $Q_y X$. It achieves the goal of the fairness task defined above if we let $\widetilde{X} = (I - Q)X$. □

**Remark A.1.** *We should emphasize that in the above theorem, the random vectors $X$, $Y$, and $Z$ are defined on the Euclidean space $\mathbb{R}^{p_1}$, $\mathbb{R}^{p_2}$ and $\mathbb{R}^{p_3}$ respectively. For each random variable, taking $X$ as an example, the sample space is defined as $\Omega = \mathcal{B}(\mathbb{R}^{p_1})$, which is Borel set generated by all open set on $\mathbb{R}^{p_1}$, and the $\sigma$-algebra $\Sigma$ is generated by $\Omega$, i.e. $\Sigma = \sigma(\Omega)$. In this way, for any measurable function $f$ satisfying the sample space of $f(X)$ is included in the sample space of $X$, we have $\sigma(f(X)) \subset \sigma(X)$, and thus the desired properties of conditional independence hold in the proof*

### A.8 EFFECT OF $q$

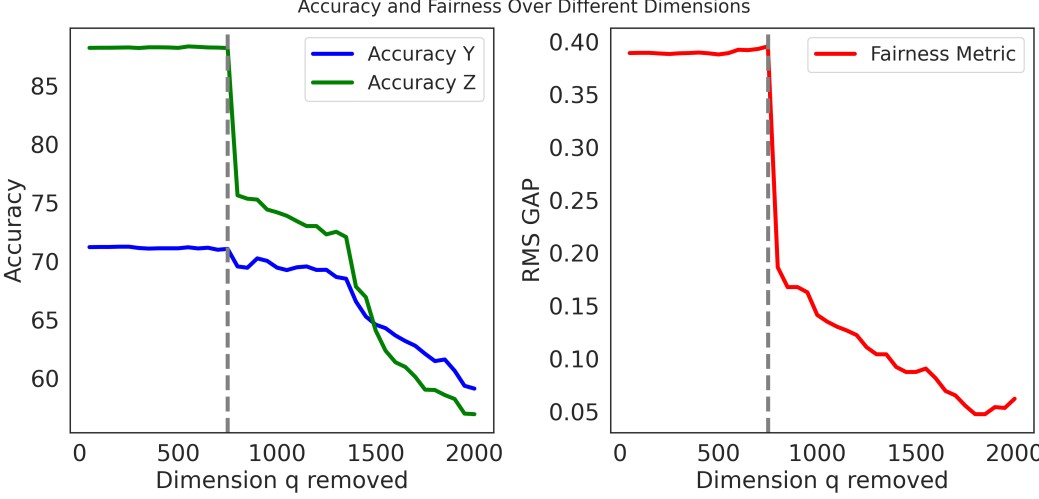

Figure 2: Trends of accuracy and GAP for MOJI data with number of dimension $q$ removed.

It is important to note that the debiasing procedure may distort the relevant concepts or key information, denoted by $Y$. Generally, as $q$ increases, both the sensitive information $Z$ and part of the target information $Y$ are excluded from the debiased representation. This occurs due to the intersection of the subspaces spanned by $Z$ and $Y$. Consequently, a rise in $q$ leads to a simultaneous reduction in accuracy and the gap, illustrating a delicate equilibrium and trade-off between targeting and debiasing performance.

### A.9 LIMITATIONS

All our result is based on the English dataset, as there is a lack of benchmark of fairness in other languages. Also, we only consider the transformation under a linear framework, where we aim to find the projection matrix $P$. However, the estimation procedure for the central subspace $\mathcal{S}_{Z|X}$ has

been well developed and can find nonlinear transformation $g$, which we leave for future exploration. Also, for the SEAT evaluation, there are some researchers point out that SEAT sometimes not able to detect the bias inside the language model. But compared with other debiasing studies that only report on SEAT, we test our method on much more comprehensive experiments.

## A.10 ETHICS STATEMENT

Our research is fundamentally methodological in nature, focusing on the development of strategies to mitigate biases in NLP. We have taken careful measures to ensure that our work adheres to recognized ethical guidelines. For all evaluations related to bias and fairness, we have strictly followed established protocols, utilizing well-known tasks to evaluate biases related to gender, religion, and race. It is important to clarify that our use of these tasks is for analytical purposes only, with the sole intention of understanding and minimizing the biases present in AI systems. Our goal is to promote fairness and inclusivity in AI, and we firmly advocate for the respectful and unbiased treatment of all individuals, irrespective of their gender, religion, or race.

## A.11 REPRODUCIBILITY

Hyperparameter tuning: For our method, the main hyperparameter is the $q$: the number of directions we want to project. We use regular grid search to find the best hyperparameter. For classifiers mentioned in Algorithm 1, we use the logistic classifier in sklearn.

Computational detail: We conduct all our experiments on an Ubuntu Server with CPU AMD Ryzen Threadripper 3990X 64-Core Processor and 256G RAM. Since our experiments do not need many computational resources (no retraining or fine-tuning), no GPU is needed.

Baseline results: Most of the baseline results are from recently published papers of well-known conferences. In static embedding evaluation, the INLP results are calculated by our code using the embedding they provided, which has a slightly better result than they reported in their paper.

