# OpenReview forum: "Bridging Debiasing Tasks with Sufficient Projection: A General Theoretical Framework for Vector Representations"
_ICLR.cc/2024/Conference — ICLR 2024 Conference Withdrawn Submission_

### Official Review · Reviewer_ZSuS · 2023-10-29

**Soundness:** 1 poor
**Presentation:** 3 good
**Contribution:** 2 fair
**Rating:** 3
**Confidence:** 3

**Summary:**

This paper applies the dimensionality reduction method of pooled marginal slicing (PMS) to find a linear subspace of word embeddings to project away information pertaining to the sensitive attribute for debiasing, which adds to a long line of research on mitigating (gender) bias in word embedding models.  The authors demonstrate the utility of the method on several benchmark tasks/datasets, and provide a theoretical justification for the proposed method.

**Strengths:**

- The paper is well-written and easy-to-follow.
- The proposed method is competitive with existing methods.

**Weaknesses:**

1. Description of hyperparameters is missing.  How was $H$ decided, and $q$ chosen, in algorithm 1?  I expect a tradeoff between accuracy and fairness for different settings of $q$, and it would be nice to include such a plot it in the paper.

2. The theoretical justifications/contributions—which is the main contribution of the present work highlighted by the authors— are very weak.

	- The *theory* relies on the strong assumption that "there exists a full rank matrix $B \in \mathbb R^{p_1\times q}$, such that $Z\perp X \mid B^\top X$".  Is there a theoretical discussion on the scenario where this assumption is violated?  How would the proposed method perform in such cases?
	- The authors lists *independence* and *separation* as the two desiderata of their method in definitions 3.1 and 3.2.  On the other hand, it is well-known that these cannot generally be simultaneously attained, see <https://fairmlbook.org/classification.html#relationships-between-criteria>.  How is it reconciled in the proposed framework?
	- The second result in theorem 6.1 is, in my opinion, too weak and not really meaningful.  If $X\perp Y\mid Q_y X$ and $\mathrm{span}(Q_y)\subseteq \mathrm{span}(Q)$, then $\widetilde X = (I-Q)X\perp Y$, i.e., the fair representation is useless for predicting $Y$ (related to the incompatibility between *independence* and *separation*).  In this regard, the first contribution claimed in section 1 of bridging the debiasing and fairness tasks does not hold up.

3. Some design choices are not (theoretically) justified.

	- In section 4.2, when $Z$ is categorical, why is there a need of learning a classifier instead of just using the one-hot encoding?

In summary, I think the authors has demonstrated that PMS is effective at debiasing word embeddings at least when compared to existing methods, although, I have my reservations on the practical usefulness of performing linear projection for debiasing word embeddings.  In particular, for text classification, there are more powerful methods (with theoretical guarantees) at achieving EO than performing linear debiasing (see follow-ups of Hardt et al. (2016)).  Instead, I would be more convinced if the author can demonstrate improved performance on WinoBias or WinoGender NLP tasks, for which bias mitigation methods for general classification tasks would not apply.

On the other hand, the theoretical discussions, as mentioned above, seems lacking if not problematic, and this weakens the overall contribution of the present work—is there anything new besides showing that PMS can give good empirical debiasing performance on the benchmark datasets?

**Questions:**

- What does *evenly* mean in "We first split all the words evenly into two classes by calculating the cosine similarity between..."?

---

> ### Author Response · Authors · 2023-11-17
>
> Thank you for the constructive feedback, which has been very helpful in enhancing the quality of our paper. We respond below to your questions and concerns:
>
> **W1**: H is chosen as a fixed number H=100 for all experiments. q is chosen by the validation set in BIOS and MOJI classification tasks. As q increases, both the sensitive information Z and a portion of the target information Y will be removed from the debiased representation. For a comprehensive visual depiction of this relationship, please refer to A.8. in the updated version.
>
> **W2**:
> 1. Firstly, our use of the linearity assumption aligns with standard practice in projection-based methods, and is widely considered a mild condition from a statistical standpoint. This assumption, which encompasses models of the form described in Eq. 1, is a relatively general case in statistical analysis.
>
> 2. Sorry for the confusion, you might misunderstand us.  It suggests that debiasing and fairness goals are not concurrently attainable within **the same classification task**. However, our work distinguishes itself by considering different tasks for achieving debiasing and fairness objectives separately, rather than attempting to fulfill both within a single task. In addition, the two tasks settings are totally different, the debias tasks in our paper are unsupervised learning tasks while the fairness tasks are supervised classification tasks.
>
> 3. First, the purpose of introducing these two definitions is to elucidate the statistical link between them, demonstrating how both can be encompassed within the conditional independence framework. Also, our work focuses on achieving debiasing and fairness objectives separately for different tasks. The second results in Theorem just want to show when both objectives can be achieved on the same task. Therefore, the concern you raised does not undermine the significance of our contribution.
>
> **W3**: The sensitive attribute Z does not have to be continuous, the main idea behind SIR is to effectively capture the relationship between the Z and the X. SIR can identify the linear combinations of predictors that are most informative about the response variable. However, the relationship between predictors and the response might be non-linear (The f in Eq. 1 can be non-linear). By partitioning the response variable, SIR can better approximate these non-linear relationships. Each slice of the response variable can be viewed as a local region where the relationship between the predictors and the response might be more linear. Therefore, the finer the interval partitioning, the better it can capture the association between X and Z. When Z is a discrete variable, the maximum number of partitions for the interval is limited to the number of values that the variable can take. This limitation prevents SIR from effectively approximating the true subspace. Therefore, we first train a classifier to output continuous values, allowing us to obtain a more precise projection through finer partitioning.
>
> **All**: The projection-based method we propose has been widely acknowledged in top-tier conferences [list references]. Within the linear projection framework, while other research primarily focuses on developing methods to obtain a useful projection for their intended goals, our approach stands out by providing an optimal projection grounded in the principle of conditional independence. This involves effectively removing the minimal subspace that contains sensitive information. To our knowledge, this theoretical attribute has not been discussed or achieved in other projection-based studies. Also, we wish to emphasize that our algorithm operates within a considerably more challenging setting compared to most of the baselines. Our key objective is to present a general framework capable of addressing both intrinsic and extrinsic tasks. In contrast, the majority of existing baselines tend to focus exclusively on either rectifying intrinsic biases or optimizing a specific downstream task. For WinoBias tasks, we indeed considered them, but the reason why we did not include them is that their evaluation metric of bias is different from our algorithm's goal, we aim to achieve fairness with the definition of conditional independence.
>
> **Q1**: Sorry for the confusion. To clarify, our methodology involves initially calculating the bias score for each word embedding, which is measured by the projection onto the gender direction. From these scores, we select words in two categories: those with the top K highest scores, representing male-associated words, and those with the top K lowest scores, representing female-associated words. These selected words then form our input data. In our practical application, we have chosen K=9000 as the number for this selection process.
>
> By clarifying this misunderstanding and addressing the other questions, I hope you might reconsider our work and, if possible, adjust the score accordingly, thank you!

---

> ### Comment · Reviewer_ZSuS · 2023-11-22
>
> I thank the authors for the response.  However, my concerns remain:
>
> W2.
>
> - Although this assumption is standard in statistics, this may not be true in practice.  As a result, the proposed method will only work if this assumption holds.  However, there are methods that do not generally require such assumptions, e.g., adversarial debiasing.
>
> - The two goals are indeed different.  The proposed algorithm only handles independence.  I fail to see how it handles separation unless with the stronger assumption in theorem 6.1.
>
> - Still, the assumption required by the discussions in Section 6, which is that $X\perp Y\mid Q_y X$ is too strong to the point that it trivializes the result.
>
> W3.  Then the need for having to train a predictor for $Z$ to get soft labeling rather than directly using the one-hot encoding is due to the linearity assumption required by the algorithm/framework.  This seems ad-hoc to me, and not very principled.
>
> All.  Being unable to handle general NLP tasks severely limits the scope and applicability of the proposed framework/algorithm.  If all I care about is text classification, then why not use the general-purpose and far more efficient fair post-processing methods in, e.g., https://fairlearn.org/, as mentioned by Reviewer PbJ8?
>
> While I appreciate the authors' detailed response, but as it stands, I am keeping my score.

---

### Official Review · Reviewer_PbJ8 · 2023-10-30

**Soundness:** 3 good
**Presentation:** 2 fair
**Contribution:** 2 fair
**Rating:** 3
**Confidence:** 3

**Summary:**

This paper provides a debiasing method (SUP) that identifies minimal subspace correlated with sensitive attribute and project vectors against them. Under the linearity assumption, authors show that the suggested method can recover fair representation consistently. Authors show that the suggested method is superior to previous methods via evaluation using intrinsic metrics (word embedding bias) and extrinsic metrics (fair text classification performance).

**Strengths:**

- This method is applicable to continuous sensitive variables.
- Using a traditional tool (Pooled Marginal Slicing), the suggested method can provide theoretical guarantee under some conditions.
- Using intrinsic, extrinsic metrics, evaluation is conducted in two levels, supporting the connection between debiasing and fairness.

**Weaknesses:**

- While the method can be applicable to any tasks that use vector representations, experiments are limited to NLP tasks.
- Most of theoretical results are trivial from the linearity assumption and method.

**Questions:**

- Can this result be extended to other modalities? (e.g. fair image classification) While applications to other modalities are implied in the discussion, still I wonder if it is effective in other modalities as well.
- Can the asymptotic result for PMS in this setting be provided? I am curious about the sample complexity and how the number of partitions affect. Also, it would be nice if it is possible to reveal how the estimation error propagates to bias with some quantifiable measures.
- Can additional comparison with fair post processing methods be conducted? (e.g. https://fairlearn.org/)

---

> ### Author Response · Authors · 2023-11-17
>
> Dear reviewer, thank you for the constructive feedback, which has been very helpful in enhancing the quality of our paper. We respond below to your questions and concerns and hope you can reconsider our work and, if possible, adjust the score accordingly, thank you!:
>
> ---
>
> **W1**: The scope of our study was deliberately chosen to provide a deep and thorough exploration of NLP-related biases, which, we believe, is a significant area of concern in itself.
> For the representation, we aim at different representations in NLP ranging from static embedding, contextualized embedding, and representations in the classification tasks. If you could find the recent research and baseline papers that work in the area of NLP [1-4], we have already made great progress in reducing the bias in NLP tasks and have conducted very comprehensive NLP bias experiments(as recognized by Reviewer acJ7) and also provide unique theoretical analysis and bridging. We hope that by doing a more fair comparison would you reconsider the contributions we made.
>
> [1] Shauli Ravfogel, Yanai Elazar, Hila Gonen, Michael Twiton, and Yoav Goldberg. Null it out:
> Guarding protected attributes by iterative nullspace projection. In Proceedings of the 58th Annual
> Meeting of the Association for Computational Linguistics, pages 7237–7256, 2020.
>
> [2] Shauli Ravfogel, Michael Twiton, Yoav Goldberg, and Ryan D Cotterell. Linear adversarial concept erasure. In International Conference on Machine Learning, pages 18400–18421. PMLR, 2022a.
>
> [3]Yue Guo, Yi Yang, and Ahmed Abbasi. Auto-debias: Debiasing masked language models with
> automated biased prompts. In Proceedings of the 60th Annual Meeting of the Association for
> Computational Linguistics (Volume 1: Long Papers), pages 1012–1023, 2022.
>
> [4]Lei Ding, Dengdeng Yu, Jinhan Xie, Wenxing Guo, Shenggang Hu, Meichen Liu, Linglong Kong,
> Hongsheng Dai, Yanchun Bao, and Bei Jiang. Word embeddings via causal inference: Gender
> bias reducing and semantic information preserving. In Proceedings of the AAAI Conference on
> Artificial Intelligence, volume 36, pages 11864–11872, 2022.
>
>
> **W2**: Under the linearity assumption, our approach uniquely identifies and eliminates the minimal subspace that contains the sensitive attribute, ensuring minimal loss of relevant target information. This theoretical property is significant and non-trivial, distinguishing our method from existing projection-based techniques in the literature, none of which have achieved this specific capability and it is also demonstrated through the various experiments.
>
> **Q1**:
>  We understand your perspective on extending our work to another modal. We are now working on the experiment of fair image classification and we will provide you with the experiment result in the next rebuttal message.
>
> **Q2**: The property of multivariate estimator can be guaranteed by proposition 4.1. For details you can refer to the following paper:
>
> Sliced inverse regression for multivariate response regression. Journal of statistical planning and inference 139, no. 8 (2009): 2656-2664.
>
> Proposition 1 of this paper establishes that the central subspace of can be decomposed as the direct sum of the central subspaces of the individual coordinates. Moving forward, Proposition 2 underscores that the space spanned by the PMS estimator (also defined in eq (2) in this paper) is indeed equal to the central subspace under mild assumptions. This essentially signifies that our estimator retains the essential properties for subspace estimation. The finer the interval partitioning, the better it can capture the central subspace. In practice, we take $H=100$.
>
> **Q3**: The fair postprocessing methods you provide only apply to the text classification task(extrinsic metrics), but it is not able to be applied to the intrinsic metrics at all. The key contribution of our method is that we provide a framework that works on both tasks. It is also not a fair comparison.

---

> > ### Author Response · Authors · 2023-11-19
> >
> > We conducted our experiment on fairness image classification using the CelebA dataset, which consists of 202,599 human face images with 40 features per image [1]. We choose Smiling and Young as the predicted labels for binary class classification. For each predicted label, we choose gender (male and female), hair color (black hair and others), and skin color (painful skin and others) as the bias-sensitive attributes. We used ResNet-50 to extract the vector representation of the CelebA dataset. Then we applied our method to obtain the debiased representation and followed the same settings in our text classification tasks to obtain the testing Accuracy and Gap between biased and debiased results.
> > The experiments show that our method also works on the fair image classification task.
> >
> > | Target   | Sensitive | Original Acc | Original Gap | Debiased Acc | Debiased Gap |
> > |----------|-----------|--------------|--------------|--------------|--------------|
> > | Young    | Gender    | 84.7         | 27.5         | 80.0         | 2.1          |
> > | Young    | Hair      | 84.7         | 19.4         | 81.3         | 10.5         |
> > | Young    | Skin      | 84.7         | 4.4          | 83.4         | 3.8          |
> > | Smiling  | Gender    | 79.9         | 7.4          | 74.5         | 5.7          |
> > | Smiling  | Hair      | 79.9         | 0.7          | 78.8         | 0.1          |
> > | Smiling  | Skin      | 79.9         | 8.6          | 79.5         | 7.9          |
> >
> >
> > [1] Ziwei Liu, Ping Luo, Xiaogang Wang, and Xiaoou Tang. Deep learning face attributes in the wild. In
> > Proceedings of International Conference on Computer Vision (ICCV), December 2015.

---

> ### Comment · Reviewer_PbJ8 · 2023-11-19
>
> Thank you for providing the detailed response and updated results.
>
> ```jsx
> Q3: The fair postprocessing methods you provide only apply to the text classification task(extrinsic metrics), but it is not able to be applied to the intrinsic metrics at all. The key contribution of our method is that we provide a framework that works on both tasks. It is also not a fair comparison.
> ```
>
> Regarding Q3, I am not convinced about why it is not a fair comparison. While I understand the fair postprocessing methods are tailored to extrinsic metrics and may not directly apply to intrinsic metrics, I believe the evaluation can still offer valuable insights into the effectiveness of the proposed method in the context of extrinsic metrics. I would appreciate further clarification on why you find the comparison unfair, as this will help me better understand your perspective.
>
> While I find theoretical properties of the suggested method interesting, I think the theoretical contribution of this paper itself is a simple extension or combination of existing results. Also, authors are highlighting the point that the suggested method is effective in both intrinsic and extrinsic metrics, but in practice any embedding debias method improves both of intrinsic and extrinsic metrics. To enhance the manuscript, I recommend the authors provide a more nuanced discussion of the unique theoretical contributions that distinguish their approach from existing results.
>
> While I am considering a marginal adjustment to my evaluation score considering the updated results provide an additional support for the generality of the proposed method, I maintain that substantial revisions are necessary for this manuscript to reach its full potential. Specifically, the paper could benefit from:
>
> - Theoretical derivation on the suboptimality of other debias methods: A more in-depth exploration of the suboptimality of other debias methods in comparison to the proposed method. Clarifying why estimating the "minimal" subspace is crucial and illustrating its impact on bias and fairness will strengthen the theoretical foundation.
> - Embedding debias in large language models: Further exploration of the proposed method's applicability to token embedding debiasing, especially in the context of large language models. An analysis of its potential effectiveness in generating fair answers for such models would significantly contribute to the paper's depth and applicability.

---

### Official Review · Reviewer_acJ7 · 2023-10-30

**Soundness:** 4 excellent
**Presentation:** 4 excellent
**Contribution:** 3 good
**Rating:** 5
**Confidence:** 4

**Summary:**

This paper provides a method for representation debiasing - the task of removing sensitive information from a learned representation post-hoc. The proposed method takes a projection-based approach, leveraging two existing approaches (SIR and PMS) to learn covariance matrices and taking their eigenvectors. Empirically, they run a range of experiments on word embeddings comparing to baselines demonstrating improved performance on metrics that look at both bias in feature space and classification.

**Strengths:**

- clear presentation
- thorough experimental section: I appreciated the quantity of experiments and baselines. The empirical component of this is I think a pretty significant contribution, as I don't think I've seen so many debiasing methods lined up side by side like this
- empirical results are strong and if they hold in general would be a nice improvement over other approaches in the space

**Weaknesses:**

- I think the "Debias" and "Fairness" terminology introduced in Defns 3.1, 3.2 is misaligned with how these words are usually used: I would consider these tasks to both be debiasing and both be fairness tasks. This is a simple fix but I think a very important one - I think there are many other ways to map these concepts onto notions of fairness/bias which make more sense (e.g. they correspond to different fairness metrics - demographic parity and equalized odds - I'm not saying this is the way authors have to go but it's an option)
- I think the contribution of the empirical section is nice - however I'm unsure of exactly how far the methodological contribution goes. It seems that the core of this method is in the regression approaches SIR + PMS. It would be good to know a bit more about why these approaches were chosen and why they are superior to other projection approaches, including those that already lean on eigenvalues (e.g. PCA-based approaches)
- Additionally, given how core SIR/PMS are to this method, I think they could use some more explanation: for instance, why is Z partitioned into H intervals? why does PMS require continuous inputs? how well does it work to first train the classifier f for its probability outputs?
- In general, I'm left a little bit unclear from Algo 1 how the "debiased" and "fair" representations, as they are referred to, are computed differently. It seems like everything specified here is for the X \indep Z setting, rather than the X \indep Z \| Y setting
- I find myself a little confused by some of the experiments in Sec 5.1, it seems like some details are bit short: for instance, I don't quite understand terms like "bias-by-projection" in Correlation or "original bias" in Profession Words. How are the top 500 male/female words in Clustering chosen?
- Also missing a few details in Sec 5.4: what model is used for prediction? is Time in seconds? specify more clearly what the predicted labels are
- I found Thm 6.1 a little confusing: 1) should the last statement be (I - Q_y)X? 2) is the assumption backwards - should span(Q_y) >= span(Q)? I looked at the proof and didn't it very enlightening, I think it could use a little more detail

Smaller comments:
- Assumption 3.3 + Eq. (1): it seems like there's a contradiction here - Ass'n 3.3 says that Z is perfectly predictable from a linear function of X, then (1) says there's a noise term included as well

**Questions:**

- Clarification: why would we need an intersection of sufficient dimension reduction subspace? in what cases would they differ from each other?
- Should clarify: is the output of PMS still minimal? it seems like the dimension-wise aggregation might result in a space with redundancies, but I could be wrong
- In general, a number of experimental details seem to be missing: for instance, how is q chosen? what about H?
- Interested to see that SUP improves over Glove in Table 2, left - I think this merits further exploration
- I'd also like to see a comparison in Sec 7 to a PCA-based approach since that one also seems similar and solves directly for the debiasing direction

---

> ### Author Response · Authors · 2023-11-17
>
> Thank you for the constructive feedback, which has been very helpful in enhancing the quality of our paper. We respond below to your questions and concerns:
>
> **W1**: Indeed, the reason why we use the term debias is that all those fairness metrics(DP, EO) are related to the supervised learning task, which has the target label Y. While for the word embedding debias(and other intrinsic tasks), they are more of unsupervised tasks(do not have the target label Y). To distinguish these two tasks, we are using two different notions to denote them.
>
> **W2**: The main methodology contribution is mainly twofold:
> 1. We form both debiasing and fairness problems from a statistical independence perspective, this is the first theoretical definition of debias in NLP.
> 2. As we discuss in the third paragraph of the introduction, there remains a significant research gap in understanding how to find the connection between these two kinds of tasks, and from our statistical independence perspective, we are able to build a theoretical linkage and experimentally show our effectiveness.
>
> Unlike other projection-based methods, our SIR+PMS approach identifies a projection that targets only the minimal subspace containing the sensitive attribute. This specificity not only enhances efficiency but also preserves a greater number of useful features in the representation, striking a balance between eliminating bias and maintaining data integrity.
>
> **W3**: The sensitive attribute Z does not have to be continuous, the main idea behind SIR is to effectively capture the relationship between the Z and the X. SIR can identify the linear combinations of predictors that are most informative about the response variable. However, the relationship between predictors and the response might be non-linear (The f in Eq. 1 can be non-linear). By partitioning the response variable, SIR can better approximate these non-linear relationships. Each slice of the response variable can be viewed as a local region where the relationship between the predictors and the response might be more linear. Therefore, the finer the interval partitioning, the better it can capture the association between X and Z. When Z is a discrete variable, the maximum number of partitions for the interval is limited to the number of values that the variable can take. This limitation prevents SIR from effectively approximating the true subspace. Therefore, we first train a classifier to output continuous values, allowing us to obtain a more precise projection through finer partitioning.
>
> **W4**: Yes, Algorithm 1 is specifically designed to handle variables X and Z, with the primary objective of eliminating the sensitive attribute present in X. This forms a unified version of our method, which applies to both debiasing and fairness tasks. It's important to note that debiasing tasks do not involve the target attribute Y, a key distinction of fairness tasks.
>
> **W5**: Thank you for pointing that out, in the submitted version we already ran out of spaces, so we kept the description concise, we are sorry for the confusion and we will add more details about those experiments in the appendix. All of the experiments in section 5.1 are from [1] and have been widely used in NLP bias research and the baseline papers. We follow exactly the same code and evaluation methods from them.
>
> All the confusion is related to the bias by projection, which is by taking the projection(projection of one vector to another vector) of a word embedding to the gender direction vector(he-she) and it is used to measure the gender bias of each word embedding.
>
> So the correlation task, it is using projection value to see how much it correlated to the neighbor result. The task is to evaluate bias in words that are close to socially marked feminine words, for example, “nurse” being close to “receptionist”, “caregiver” and “teacher”. The original bias means the bias in the original embedding before the debias, the Profession Words task is a sub-task of the correlation task that focuses only on the professional words. Again, the top 500 words are chosen by the projection and chose the top 500 and the last 500 words.
>
> [1] Gonen, Hila, and Yoav Goldberg. "Lipstick on a pig: Debiasing methods cover up systematic gender biases in word embeddings but do not remove them." arXiv preprint arXiv:1903.03862 (2019).
>
> **W6**: The logistic regression model serves as our chosen prediction method, a choice consistent with previous literature in this field. We measure the model's performance time in seconds. For the BIOS dataset, the model classifies a given biography text into 28 different professions labels(like "accountant", "professor", etc.), while for the MOJI dataset, it classifies emotions into positive or negative categories.

---

> ### Author Response · Authors · 2023-11-17
>
> **W7**: The statement in the Theorem is correct. Furthermore, the second result of our theorem takes an additional step, building upon the premise of an already debiased representation. It specifically demonstrates the circumstances under which a representation can simultaneously achieve both debiasing and fairness objectives within the same task.
>
> **Comment**: Apologies for any confusion earlier. To clarify, when we mention that it can be 'perfectly predicted,' we refer to the underlying model that utilizes only the projection of X on q directions as its input. This model setup includes a noise term, akin to the inclusion of a noise term in a linear regression model. Thus, the term 'perfectly' in this context doesn't imply the absence of noise, but rather that the model's input is exclusively based on this specific projection of X.
>
> **Q1**: In the realm of conditional independence, there exists an infinite number of subspaces that can fulfill this criterion. However, our focus is on identifying the minimal subspace, which essentially represents the intersection of all such subspaces. For instance, consider a situation where Span$\{v_1, v_2\}$ constitutes the central (or minimal) subspace. In this case, a larger subspace like Span$\{v_1, v_2, v_3\}$ would also satisfy the condition of conditional independence. This implies that any space capable of achieving conditional independence must inherently contain the central subspace.
>
> **Q2**: The property of multivariate estimator can be guaranteed by proposition 4.1. For details you can refer to the following paper:
>
> Sliced inverse regression for multivariate response regression. Journal of statistical planning and inference 139, no. 8 (2009): 2656-2664.
>
> Proposition 1 of this paper establishes that the central subspace of can be decomposed as the direct sum of the central subspaces of the individual coordinates. Moving forward, Proposition 2 underscores that the space spanned by the PMS estimator (also defined in eq (2) in this paper) is indeed equal to the central subspace under mild assumptions. This essentially signifies that our estimator retains the essential properties for subspace estimation.
>
> **Q3**: $H$ is chosen as a fixed number $H=100$ for all experiments. $q$ is chosen by the validation set in BIOS and MOJI classification tasks.
>
> **Q4**: For Table 2, left, the main goal is to show that our method does not destroy(no decrease of the wordsim tasks) the original semantic information, and as you see in the table most of the result is only slightly increased. The phenomenon of increasing the word similarity tasks is also widely seen in many of the baseline papers, but indeed, we need to have further exploration on this.
>
> **Q5**: The PCA-based approach does not ensure that the estimated subspace, which contains sensitive information, is the smallest possible. This is a key distinction from our SUP method, which is specifically designed to identify the minimal such subspace. We are open to a more detailed comparison with any specific PCA-based approach you might have in mind, to further elucidate the differences and advantages of our method.
>
> ---
>
> By clarifying this misunderstanding and addressing the other questions, I hope you might reconsider our work and, if possible, adjust the score accordingly, thank you!.

---

> > ### Comment · Reviewer_acJ7 · 2023-11-20
> > **Response 2**
> >
> > W7: I believe the theorem is correct - my feedback is that it could use some more explication in the proof since it isn't obvious to me.
> >
> > Re other comments: thanks for the clarifications and information - I think a lot of this information should be in the paper. At the moment, given the response and my continued lack of clarity around contributions and framing, I'm not going to increase my score, but I will take this rebuttal into account in discussion with other reviewers.

---

> ### Comment · Reviewer_acJ7 · 2023-11-20
> **Response 1**
>
> Thanks for the rebuttal - responding point by point here:
>
> W1: I disagree with this characterization - both the "debias" and "fairness" metrics as you introduce them are about representation distributions, neither are about supervised classification. Further, a metric like DP is actually unrelated to a label Y so I'm not sure I agree with that distinction either.
>
> W2: I disagree that this is the first theoretical definition of representation debiasing - for instance, see "Learning Controllable Fair Representations" for a similar approach based on minimizing mutual information (there are many other papers you could look at, this is just the first one that came to mind). It's possible this is the first explicitly in the NLP space, but I'm not sure how the definition would differ for the NLP case. I agree there's a gap in understanding the impact of feature debiasing on fair classification, with some results in the space but not fully explaining all behavior - there's some theoretical linkage here in Sec. 6 but I think there needs to be more explication of how this differs from other projection methods for readers to understand the contribution. I agree with the experimental contribution.
>
> W4: I guess I'm still missing some understanding here - is SUP intended to apply to both "debias" and "fairness" tasks? if so, I'm a little unsure why the approach wouldn't vary for both. If not, then is there a version which is used for "fairness" tasks?
>
> W5: I'm familiar with the general approaches and type of experiment that is run in this area - my feedback was that some of these terms and procedures are not explicitly defined in the paper, and as such it's hard to understand what exactly was done at points.
>
> W6: okay - if this information isn't in the paper I think it should be, but I may have missed it